# Health care providers' knowledge and associated factors towards the components of post abortion care service in hospitals of Tigray, Northern Ethiopia

Gidey Goitom[1]*, Balem Dimtsu[1], Abera Haftu[1], Solomon Weldemariam Gebrehiwot[1], Gebru Hailu Redae[2]

1 Department of Midwifery, College of Health Sciences, Mekelle University, Mekelle, Ethiopia,
2 Department of Environmental Health Science, College of Health Sciences, Mekelle University, Mekelle, Ethiopia

* gideygoitom34@gmail.com

## Abstract

In Ethiopia, a significant proportion of women face barriers to accessing family planning services, with nearly 48% not receiving family planning counseling and approximately 35.2% receiving no counseling at all. This gap in provider knowledge contributes to a high rate of repeat abortions, which stands at 33.6%, underscoring the urgent need for effective post-abortion care. The main factor behind this issue is the knowledge gap among healthcare providers, which affects their ability to offer comprehensive family planning service. Addressing this gap is essential for improving reproductive health outcomes and reducing the incidence of abortion complication in the country. Therefore, the aim of this study is to assess knowledge of healthcare providers regarding post-abortion care components and the associated factors influencing this knowledge in Tigray, Northern Ethiopia, in 2019. Hospital-based cross-sectional study employed using face-to-face interviews from May to August 2019. A total of 224 healthcare providers, including midwives, nurses, doctors, emergency surgeons, and health officers, were selected from seven randomly chosen general and referral hospitals in Tigray. Data were entered, cleaned, and analyzed using SPSS version 20. Both binary and multivariable logistic regression analyses were conducted to explore the relationship between various independent variables and providers' knowledge of components of post-abortion care. The study found that only 25.9% of healthcare providers had adequate knowledge about the components of post-abortion care. Regarding the components of post abortion care, family planning and community- care provider partnerships were recognized by the 91.1% and 36.2% of participants respectively as a components of post abortion care. Providers who used post-abortion care guidelines were three times more likely to be knowledgeable (AOR = 3.424 [1.09–10.72]) compared to those who did not. Similarly, those who received training were four times more knowledgeable (AOR = 4.102

**Data availability statement:** All data analysed during this study are included in this manuscript as supplementary information files.

**Funding:** Mekelle University College of Health Sciences funded only for the data collection to GG. with a grant number of:CRPO/CHS/SM/011/14. However, the funder did not have a role in design, analysis and a decision to submit the manuscript for publication.

**Competing interests:** The authors have declared that no competing interests exist.

[1.71–9.79]). Providers who managed five or more cases per day were also four times more knowledgeable (AOR = 4.348 [1.57–11.97]) compared to those managed less than five cases, and those working in referral hospitals demonstrated four times better knowledge compared to their counterparts in general hospitals (AOR = 4.332 [1.82–10.27]). The study highlights a significant gap in healthcare providers' knowledge on components of post-abortion care. To address this, regular in-service training and adherence to national post-abortion care guidelines are strongly recommended.

## Background

Abortion remains one of the leading causes of maternal mortality worldwide, contributing nearly 25 million unsafe abortions occur annually, one of the critical barriers to improving post-abortion care is the knowledge gap among healthcare providers [1,2]. This gap contributes to 520/100,000 maternal mortality annually due to unsafe abortion [3]. Each year, around 75 million women require post-abortion care (PAC) [4]. Providing contraceptive counseling and PAC services during these visits is crucial for reducing repeat abortions [5,6].

Research indicates that women in many countries still lack access to essential post-abortion services or face mistreatment when seeking care [7]. A study conducted in Addis Ababa, Nigeria, and Pakistan revealed varying levels of healthcare providers' knowledge regarding Comprehensive PAC, with knowledge rates of 53.1%, 75.5%, and 50%, respectively [7–9]. In 2002, the PAC Consortium at Chapel Hill introduced a model with five essential components to ensure high-quality and sustainable PAC services [10]. These components are; 1. Emergency treatments for incomplete abortion using manual vacuum aspiration (MVA) or Misoprostol to remove retained products of conception [11]. If healthcare providers are not adequately knowledgeable in these emergency procedures, the risk of incomplete treatment increases, leading to complications such as infection, hemorrhage, or uterine perforation. Inadequate or improper emergency treatment can lead to severe health risks for the patient, including sepsis, shock, and even death [12]; 2. Counseling to address the emotional and physical health needs of women to address the emotional and psychological needs of women who have undergone abortion, as well as to ensure their physical recovery is closely monitored [13]. Healthcare providers who lack training in emotional and psychological support may overlook or inadequately address the mental health of patients. This can lead to anxiety, depression, or post-traumatic stress, especially in cultures where abortion is stigmatized [14]; 3. Post-procedure family planning counseling and services, this component ensures that women receive counseling on family planning options after the abortion procedure, along with access to effective contraception methods [15]; 4. Links with other reproductive health care providers, this involves creating a network of healthcare providers that can refer women for continued care after the abortion, including routine reproductive health checkups,

STI treatment, and further family planning support [11].; 5. Partnerships with community and service providers to prevent unwanted pregnancies and unsafe abortions [11]. Those five components are essential even one of those should not be compromised [10].

According to world health organization (WHO) guidelines, the knowledge and implementation of specific sexual and reproductive health (SRH) components are still below optimal levels, and some healthcare workers hold negative attitudes toward young people using contraceptives, often limiting adolescents' access to SRH services [16,17]. Studies in Ethiopia suggest that complications from abortion may account for up to 19.6% of maternal deaths in the country [18]. In Ethiopia nearly half (48%) of women missed family planning counseling, and in another study about 35.2% didn't receive any type of family planning counseling [19,20]. The rate of repeated abortion in Ethiopia stands at 33.6%, indicating a profound need for effective post-abortion care [21].

While components of post abortion care (CPAC) practices have become common among healthcare providers, they are not always implemented comprehensively. Often, healthcare providers may neglect one or more of the five essential CPAC. This lack of full implementation can significantly affect patient outcomes. For instance, while performing MVA is a common practice, the complete integration of all CPAC remains unlikely. few studies in developing countries assess healthcare provider's knowledge and associated factors regarding CPAC. In Ethiopia, there is a lack of documented data specifically evaluating healthcare providers' knowledge and associated factors related to the five components of PAC. This study aims to fill this gap by assessing healthcare providers' knowledge and associated factors regarding components of post-abortion care services in Tigray, Northern Ethiopia.

## Methods

### Study area, period and design

A hospital-based cross-sectional study was conducted in hospitals across Tigray Region, northern part of Ethiopia. Mekelle is the capital city of Tigray region, situated 783 kilometers from Addis Ababa. The region has 2 referral hospitals, 14 general hospitals, 25 primary hospitals, 211 health centers, 718 health posts, and 452 health workers providing abortion care services. The study took place from 1 May to 30 August 2019, in government general and referral hospitals throughout the Tigray region.

### Sample size and sampling technique

**Sample size.** The sample size was calculated using a formula for determining sample size in a single population proportion, based on several assumptions. We used a proportion of 59.3% for health providers' knowledge of MVA as part of post-abortion care in Nigeria, with a margin of error of 5% and a 95% confidence interval [8]. To account for potential non-respondents, a contingency of 10% was added to the total sample size. With these assumptions, the final calculated sample size was 224.

**Sampling technique.** From the total 16 hospitals, seven were selected (5 out of the 14 general hospitals using simple random sampling, however for the referral hospitals, we decided to include all the 2 available referral hospitals in the study). Then sample size was distributed to each hospital proportionally using probability proportional to size (PPS) technique. Finally, study participants were selected using systematic random sampling method using their list from the gynecology and obstetrician unit as sampling frame.

### Inclusion and exclusion criteria

Inclusion criteria for our study were all healthcare providers actively involved in providing abortion services at the Obstetrics and Gynecology (Obs/Gyn) ward.

Exclusion criteria, we did not set any formal exclusion criteria for this study.

## Data collection procedure and instrument/tool

The questionnaire was developed after in depth reviewing of different literatures [7,9,22]. The questionnaire comprised 4 parts namely: socio-demographic characteristics, job and setup related factors, guideline use and training, and knowledge on CPAC. A Ten trained midwives and nurses professionals (7 data collector and 3 supervisor) have been participated in the data collection. We collected the data using semi-structured face-to-face interview questionnaires.

## Data quality assurance method

The questionnaire was prepared in English version and it was translated back to Tigrigna and again to English to confirm the correctness of the translation. To assure quality of data, we provided training for two days for data collectors and supervisors about the objective of the study, data collection tool and process. Pre-test was conducted on 10% of the sample size at Wukro hospital other than the study facilities to check clarity of the tool. The results of the pre-test were used to ensure clarity of language and verify skip patterns of the questions and some modifications were made. The Supervisors oversaw interviewers daily during the whole period of data collection and checked questionnaires for completeness.

## Data processing and analysis

The hard copy data we carefully reviewed for completeness and consistency. Questionnaires were visually inspected, coded, and then entered into SPSS version 20 for data cleaning and analysis. Descriptive statistics, including frequencies and percentages, were used to summarize categorical variables, while mean and standard deviations computed to describe continuous variables. We use tables to present descriptive results for some variables.

For analysis, a bivariate logistic regression model was initially applied. Odds ratios (OR) with 95% confidence intervals (CI) were calculated to evaluate the relationship between post-abortion care service components knowledge and associated factors. Independent variables with a P-value<0.25 in the binary logistic regression were further examined using multivariable logistic regression to identify key factors while controlling for confounders. Adjusted odds ratios (AOR) with 95% confidence intervals were computed for significant predictors, with statistical significance set at P<0.05.

## Operational definitions

Good knowledge: when the participants list all the knowledge related questions of CPAC were operationalized as having good knowledge.

Poor knowledge: When the participants miss one or more knowledge related questions were operationalized as having poor knowledge. Those five components are essential even one of those should not be compromised [10].

## Ethics approval and consent to participate

Ethical clearance was obtained from institutional ethical review board of Mekelle University College of health science. Official letter of permissions was obtained from Tigray regional health bureau and selected hospitals. Written informed consent from the participants was obtained after clear explanation of the purpose of the study. Information was collected anonymously and confidentiality was assured throughout the study period. Moreover, all methods were performed in accordance with the ethical principles of the Declaration of Helsinki.

## Results

### Socio-demographic characteristics

Data were collected from a total of 224 study participants, yielding a 100% response rate. Among these participants, 130 (58%) were female. The ages of the participants ranged from 20 to 56 years, with a median age of 30.00±10 years

(IQR 10). Approximately 190 (85%) of the participants were of Tigrayan ethnicity. Around 113 (50.4%) were married, and 187 (83.5%) were Orthodox Christianity followers. Additionally, 188 (83.9%) of the participants had attained a bachelor's degree (Table 1).

### Job and setup related factors

About 132 (58.9%) of the participants had work experience ranging from 6 months to 5 years. A total of 135 participants (60.3%) were employed in general hospitals, and the majority 129 (57.6%) were Midwifery. One hundred ninety-seven (88%) of the participants expressed their motivation to read about abortion care. One hundred fifty-two respondents (67.9%) reported being supervised by their respective supervisors, with 95 (62.5%) receiving supervision on a daily basis (Table 2).

### Guideline use and training

Among the total participants, approximately 152 (67.9%) reported that their hospitals have a guideline in place, but only 99 (65.1%) actually use it. The rest of participants did not use the guideline, primarily due to being too busy. Additionally, 68.6% of the participants had received training within the two years preceding the data collection period.

### Health care providers' knowledge on the components of post-abortion care

The present study revealed that 58 (25.9%) of health providers have good knowledge of the Components of Post-Abortion Care (CPAC), indicating that they missed at least one or more key CPAC. While most participants were aware of family planning (91.1%), counseling (79%), and emergency treatment (63.8%) as core CPAC, but only 36.2% and 52.7% of participants correctly identify community and service provider partnerships and linkages to reproductive health and other health services, respectively, as core CPAC.

**Table 1. Socio-demographic characteristics of health providers at hospitals of Tigray 2019.**

| Variable | | Frequency | Percent |
|---|---|---|---|
| Age | 20-30 | 115 | 51.3 |
| | 31-40 | 74 | 33.0 |
| | >=41 | 35 | 15.6 |
| Gender | Female | 130 | 58.0 |
| | Male | 94 | 42.0 |
| Ethnicity | Tigray | 190 | 84.8 |
| | Amhara | 33 | 14.7 |
| | Oromo | 1 | 0.4 |
| Religion | Orthodox | 187 | 83.5 |
| | Muslim | 21 | 9.4 |
| | Catholic and protestant | 16 | 7.2 |
| Marital status | Married | 113 | 50.4 |
| | Single | 89 | 39.7 |
| | Widow | 11 | 4.9 |
| | Divorced | 11 | 4.9 |
| Level of Education | Degree | 188 | 83.9 |
| | Diploma | 29 | 12.9 |
| | Master and above | 7 | 3.1 |

**Table 2. Job and setup related factors among health providers at hospitals of Tigray 2019.**

| Job and setup-related factors | | Frequency | Percent |
|---|---|---|---|
| Profession type | Midwifery | 129 | 57.6 |
| | Nurse | 65 | 29.0 |
| | Medical doctor | 24 | 10.7 |
| | Emergency surgery and HO | 6 | 2.6 |
| Year of experience | 6-60month | 132 | 58.9 |
| | 61-120 month | 50 | 22.3 |
| | >=121 month | 42 | 18.8 |
| Number of professions working/day | <=3 | 155 | 69.2 |
| | >=4 | 69 | 30.8 |
| Supervision by responsible bodies | Yes | 152 | 67.9 |
| | No | 72 | 32.1 |
| How often supervised | Daily | 95 | 62.5 |
| | Weekly | 36 | 23.68 |
| | Monthly | 21 | 13.81 |
| Motivation to read on abortion related topic | Yes | 197 | 87.9 |
| | No | 27 | 12.1 |
| Level of hospital | General hospital | 135 | 60.3 |
| | Referral hospital | 89 | 39.7 |
| Contraceptive | Yes | 210 | 93.9 |
| At abortion room site | | | |
| | No | 14 | 6.3 |
| Availability of reading room | Yes | 155 | 69.2 |
| | No | 69 | 30.8 |
| Case flow per day | <=4 | 177 | 79.0 |
| | >=5 | 47 | 21.0 |

Among the participants, 25 (11.2%) could list only one component, 52 (23.2%) could list two components, 46 (20.6%) could list three, 41 (18.3%) could list four, and 58 (25.9%) were able to list all five core CPAC. Notably, two participants could not list any components (Table 3).

## Factors associated with knowledge of health care providers on the components of post abortion care (CPAC)

Both binary and multivariable logistic regression analyses were conducted to identify factors associated with healthcare providers' knowledge of the CPAC. Variables such as gender, educational level, employment in referral hospitals, receipt of training, supervision by responsible bodies, use of guidelines, availability of a reading room, case flow, on-the-job training, and the number of professions held simultaneously were initially considered, with a p-value < 0.25 in the binary logistic regression analysis. These variables were then included in the multivariable logistic regression analysis.

The multivariable analysis, after controlling for potential confounders, revealed several significant findings. Providers working in referral hospitals were four times more likely to be knowledgeable about PAC compared to those working in general hospitals (p = 0.001, AOR = 4.332 [1.827–10.271]). Those who received PAC training were also four times more likely to have good knowledge compared to those who did not receive training (p = 0.001, AOR = 4.102 [1.719–9.790]). Additionally, case flow was significantly associated with knowledge: providers who handled five or more cases per day were four times more likely to be knowledgeable than those with fewer cases (p = 0.004, AOR = 4.348 [1.579–11.971]). Furthermore, the likelihood of having good knowledge was higher among those who utilized guidelines compared to those who did not (p = 0.035, AOR = 3.424 [1.093–10.728]). These results are detailed in Table 4.

**Table 3. Health care providers' knowledge on the components of post-abortion care in hospitals of Tigray 2019.**

| Variable | Listed | Frequency | Percent |
|---|---|---|---|
| Knowledge of components of PAC | | | |
| Good | | 58 | 25.9 |
| Poor | | 166 | 74.1 |
| Emergency treatment | Yes | 143 | 63.8 |
| | No | 81 | 36.2 |
| Counseling | Yes | 177 | 79 |
| | No | 47 | 21 |
| Family planning | Yes | 204 | 91.1 |
| | No | 20 | 8.9 |
| Linkage to RH and other health services | Yes | 118 | 52.7 |
| | No | 106 | 47.3 |
| Community and service provider | No | 143 | 63.8 |
| Partnership | Yes | 81 | 36.2 |

## Discussion

The current study found that the knowledge of participants regarding the components of post-abortion care (CPAC) was 25.9%, which is notably lower compared to similar studies conducted in Addis Ababa (53.1%), South Nigeria (75.5%), Pakistan (50%), and Afghanistan (55.1%) [7–9,22]. The variation in knowledge levels across different settings can be attributed to several factors. Differences in measurement methods, such as more stringent or comprehensive criteria in our study compared to others using simpler tools, may explain some of the findings. Additionally, variations in sample sizes, study designs, and the representativeness of the populations studied could influence the results.

In this study, 45.5% of the providers had received training on PAC, aligning with the 50% training rate observed in sub-Saharan Africa [16]. However, this rate is lower compared to Nigeria and Afghanistan, where 74.1% and 70% of providers had received training, respectively [8,22]. These higher training rates in Nigeria and Afghanistan could be attributed to more robust training programs or greater efforts by local governments and NGOs to prioritize PAC education for healthcare providers, which may reflect differences in national policies or the availability of resources for training initiatives.

Conversely, the training rate in this study is higher than the 20.5% reported in Addis Ababa [7]. These discrepancies may be due to differences in sample sizes, such as the 405 health providers included in the Addis Ababa study.

Regarding knowledge of specific CPAC, 91.1% of participants recognized family planning as a core component, which is higher than the 63% reported in Pakistan and 6.4% in Nigeria [8,9]. This substantial difference in knowledge could be attributed to several factors, including the relative strength and accessibility of family planning policies in Ethiopia. In Ethiopia, family planning services are widely promoted and integrated into reproductive health programs, making them more accessible to healthcare providers and the general population.

The study found that 36.2% of participants identified community provider partnerships as a CPAC, which is higher than the 6.8% reported in Nigeria and 29% in Pakistan [8,9]. This difference may variations in the timing of the studies, differences in sample sizes, and the types of participants involved. For instance, both the studies in Nigeria and Pakistan focused primarily on mid-level healthcare providers, who may have different perceptions or levels of exposure to community partnerships compared to other healthcare professionals.

Approximately 53% of participants recognized linkage to other reproductive health services as a CPAC, which is higher than the 40% reported in Pakistan [9]. However, 36.2% of participants did not identify emergency treatment as a CPAC, consistent with the 40% found in southern Nigeria [8]. This difference could be regional healthcare policies, the integration of reproductive health services, and the level of awareness among healthcare providers.

**Table 4. Multivariable analysis showing factors associated with health providers' knowledge on components of post abortion care 2019.**

| Variable | Knowledge | | COR (95%CI) | AOR (95%) | P-value |
|---|---|---|---|---|---|
| | Good knowledge | Poor Knowledge | | | |
| | (25.9%) | (74.1%) | | | |
| Gender | | | | | |
| Male | 32(55.2%) | 62(37.3%) | 2.065(1.127-3.783) | 1.770(.776-4.036) | 0.175 |
| Female | 26(44.8%) | 104(62.7%) | 1 | 1 | |
| Level of Education | | | | | |
| Diploma | 5(8.6%) | 26(15.7%) | 1 | 1 | |
| Degree | 48(82.8%) | 135(81.3%) | 1.849(.672-5.087) | .374(.093-1.497) | 0.164 |
| Master | 5(8.6%) | 5(3.0%) | 5.087(1.086-24.897) | .362(.040-3.282) | 0.367 |
| Level of hospital | | | | | |
| General hospital | 23(39.7%) | 112(67.5%) | 1 | 1 | |
| Referral hospital | 35(60.3%) | 54(32.5%) | 3.156(1.701-5.857) | 4.332(1.827-10.271) | 0.001** |
| Training | | | | | |
| Yes | 44(75.9%) | 58(34.9%) | 5.852(2.962-11.561) | 4.102(1.719-9.790) | 0.001** |
| No | 14(24.1%) | 108(65.1%) | 1 | 1 | |
| Supervision | | | | | |
| Yes | 50(86.2%) | 102(61.4%) | 3.922(1.746-8.808) | .593(.193-1.819) | 0.360 |
| No | 8(13.8%) | 64(38.6%) | 1 | 1 | |
| Use guideline | | | | | |
| Not available | 5(8.6%) | 67(40.4%) | .585(.168-2.028) | .711(.183-2.761) | 0.622 |
| Yes | 47(81.0%) | 52(31.3%) | 7.080(2.775-18.067) | 3.424(1.093-10.728) | 0.035* |
| No | 6(10.3%) | 47(28.3%) | 1 | 1 | |
| Reading room | | | | | |
| Yes | 52(89.7%) | 103(62.0%) | 5.301(2.152-13.056) | 1.575(.500-4.959) | 0.437 |
| No | 6(10.3%) | 63(38.0%) | 1 | 1 | |
| Case flow/day | | | | | |
| <=4 | 31(53.4%) | 146(88.0%) | 1 | 1 | |
| >=5 | 27(46.6%) | 20(12.0%) | 6.358(3.170-12.754) | 4.348(1.579-11.971) | 0.004* |
| On job training | | | | | |
| Yes | 45(77.6%) | 83(50.0%) | 3.462(1.740-6.888) | 1.166(.476-2.854) | 0.737 |
| No | 13(22.4%) | 83(50.0%) | 1 | 1 | |
| Number of profession/days | | | | | |
| <=3 | 26(44.8%) | 129(77.7%) | 1 | 1 | |
| >=4 | 32(55.2%) | 37(22.3%) | 4.291(2.277-8.086) | 1.214(.510-2.888) | 0.662 |

* p value of ≤0.05 ** p value of ≤.001.

COR- crude odds ratio, AOR- adjusted odds ratio.

Providers who received training were four times more knowledgeable than those who did not, supporting findings from Afghanistan and southeastern Africa that training significantly improves knowledge [17,22]. This aligns with WHO guidelines for safe abortion and PAC, which emphasize the importance of up-to-date training for healthcare providers [17]. This suggests that current training strategies are effectively enhancing knowledge and skills.

Access to guidelines was significantly associated with better knowledge of CPAC. Providers using guidelines were three times more knowledgeable than those who did not, echoing findings from Malawi that guidelines are a crucial factor in providing quality PAC [23]. Similarly, WHO guidelines emphasize the necessity of having guidelines to ensure high-quality care [17,24].

The level of the hospital was also significantly associated with knowledge. Providers working in referral hospitals were four times more knowledgeable than those in general hospitals, a finding consistent with studies from Afghanistan [22]. This may be because referral hospitals often function as teaching hospitals, providing access to the latest knowledge and resources.

## Conclusion

The majority of the health care providers in Tigray general and referral hospitals had poor knowledge of the components of post-abortion care. Besides, about one-third of the health providers did not use guideline. Guideline use, working place, case flow and training were significant determinant factors of knowledge.

## Limitation of the study

This study was only limited in governmental hospitals of Tigray not included private clinic and since this study was cross sectional study, it is difficult to establish the causal relationship.

## Recommendation

To Tigray Regional Health Bureau:

The Tigray Regional Health Bureau should encourage the sharing of experiences and best practices between referral and general hospitals. This can help bridge knowledge gaps and promote a more uniform standard of care across different healthcare settings.

In addition, it is essential to provide updated training to all healthcare providers who offer abortion care, to ensure they remain well-informed about the latest protocols, best practices, and guidelines.

To Health Workers and Healthcare Facilities:

Healthcare providers and facilities should be encouraged to utilize the guidelines available in their clinics, ensuring that they are consistently followed in the provision of post-abortion care.

## Supporting information

**S1 Data. Data collected from participants and analyzed during this study (SPSS data).**
(SAV)

**S1 File. Questionnaire used for data collection in this study.**
(PDF)

## Acknowledgments

We would like to express our appreciation to the study participants for their valuable time and cooperation, as well as to the data collectors and supervisors for their commitment throughout the data collection process.

## Author contributions

**Conceptualization:** Gidey Goitom.

**Data curation:** Gidey Goitom.

**Formal analysis:** Gidey Goitom.

**Funding acquisition:** Gidey Goitom.

**Investigation:** Gidey Goitom.

**Methodology:** Gidey Goitom, Balem Dimtsu, Abera Haftu, Solomon Weldemariam Gebrehiwot, Gebru Hailu Redae.

**Project administration:** Gidey Goitom.

**Resources:** Gidey Goitom.

**Software:** Gidey Goitom.

**Supervision:** Gidey Goitom, Balem Dimtsu, Abera Haftu, Solomon Weldemariam Gebrehiwot.

**Validation:** Gidey Goitom, Gebru Hailu Redae.

**Visualization:** Gidey Goitom.

**Writing – original draft:** Gidey Goitom.

**Writing – review & editing:** Gidey Goitom, Balem Dimtsu, Abera Haftu, Solomon Weldemariam Gebrehiwot, Gebru Hailu Redae.

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
