## [Decision Letter · Decision Letter 0]

17 Dec 2024

PGPH-D-24-02390

Health care providers’ knowledge and associated factors towards the components of post abortion care service in hospitals of Tigray, Northern Ethiopia.

Dear Dr. Goitom,

Thank you for submitting your manuscript to PLOS Global Public Health. After careful consideration, we feel that it has merit but does not fully meet PLOS Global Public Health’s publication criteria as it currently stands. Therefore, we invite you to submit a revised version of the manuscript that addresses the points raised during the review process.

We look forward to receiving your revised manuscript.

Kind regards,

Damen Haile Mariam, MD, MPH, PhD

Academic Editor

Additional Editor Comments (if provided):

Reviewer 1:

Abstract:

- Make the abstract more specific and concise. Include only important information in the abstract and avoid jargon. For example, in the background you described about the burden associated with complications of pregnancy and child birth and included information about the magnitude of unsafe abortion which are not specific for the topic. Please try to focus on the magnitude of the problem (knowledge of HP’s on the CPAC), what is unknown about the problem (knowledge gap), and what you are going to do about it (objective).

Background:

- Please revise your background based on the above comment. It lacks specificity and coherence.

- Try to discuss about the components of post abortion care in more detail. Define each component and reason out how the knowledge gap specific to each component affect the outcome of abortion care.

- Line 54… ‘This contributes to 520/100,000 cases of maternal deaths…’ is that per women of child bearing age or cases of maternal death? Please check.

- Line 77 ‘… (CPAC) practices have become common among healthcare providers…’ how can you justify this statement? common CPAC practices but low knowledge

Methods:

- Please revise your sample size calculation. The proportion of knowledge about MVA is only one CPAC but you are trying to study the overall knowledge about the 5 components. In addition how is the

facilities and population comparable with that of the Nigerian study?

- Sampling method: out of 2 referral hospitals and 14 general hospitals, you selected 5 general hospital and 2 referral hospitals. You said you used SRS method, but how do you determine the number

of hospitals required for study?

- Regarding participant selection, since the level of knowledge of the HP’s is affected by their level of professionalism (heterogeneous population), it is better to stratify the participants based on their

profession (gynecologist, emergency surgery professional, MW, etc.) during sampling.

- What are your inclusion and exclusion criteria?

- Data collection instrument: did you use a standard questionnaire to ass knowledge?

- Why didn’t you use self-administered questionnaires rather than face to face interview? How did you controlled interviewer bias?

- To my knowledge, Health professionals in Ethiopia are trained and practice in English and they are supposed to understand English. Why did you need to translate the questionnaire to Tigrigna?

- Please provide the questionnaire as a supporting material during submission.

- Operational definitions: what is your reference to define good and poor knowledge? Is it a standard method?

Results:

- Please revise your results part. Stick to the journal’s statistical reporting guideline.

- Please avail the data set associated with the reported data in a public repository or provide as a supporting material during submission.

Discussion:

- The discussion part is very shallow and full of grammar, spelling, and punctuation error. Please try to discuss the main findings adequately based on your objectives.

- What is your recommendation based on your study findings?

- Do you think exclusion of health centers and primary hospitals as the limitation of the study in addition to private clinics?

References:

- Please go through your reference and edit.

Reviewer 2 -

Background:

- The background did not mention the comprehensive abortion care concept which is another critical milestone on abortion services.

Methods:

- The data as well looks a bit older (about 5 years old).

Discussion:

- The discussion section is also abbreviated.

Overall:

- The paper it is informative on knowledge of providers related to components of PAC.

- The criterion used for knowledge was very stringent (100% response as good versus all those less than 100% as poor).

Reviewers' comments:

Reviewer's Responses to Questions

**Comments to the Author**

1. Does this manuscript meet PLOS Global Public Health’s publication criteria ? Is the manuscript technically sound, and do the data support the conclusions? The manuscript must describe methodologically and ethically rigorous research with conclusions that are appropriately drawn based on the data presented.

Reviewer #1: Partly

Reviewer #2: Partly

2. Has the statistical analysis been performed appropriately and rigorously?

Reviewer #1: I don't know

Reviewer #2: I don't know

3. Have the authors made all data underlying the findings in their manuscript fully available (please refer to the Data Availability Statement at the start of the manuscript PDF file)?

Reviewer #1: No

Reviewer #2: No

4. Is the manuscript presented in an intelligible fashion and written in standard English?

Reviewer #1: No

Reviewer #2: No

5. Review Comments to the Author

Reviewer #1: Abstract

The abstract should be more concise and focused on the core issue: healthcare providers' knowledge of post-abortion care (PAC) components in Tigray, Northern Ethiopia. The background in the abstract should highlight the specific knowledge gap related to PAC components and the study's objective to assess this gap.

Background

The background section should provide a more detailed overview of PAC components, defining each component and explaining its significance in preventing complications and improving maternal health outcomes. The knowledge gap specific to each component should be emphasized to justify the study's relevance.

Methodology

* Sample Size Calculation: The sample size calculation should be revised to account for the multiple components of PAC. The comparability of the study population to the Nigerian study should be justified.

* Sampling Method: The rationale for selecting specific hospitals and the number of hospitals included in the study should be clarified. Stratified random sampling based on healthcare professionals' roles (gynecologist, emergency surgeon, midwife, etc.) is recommended to account for potential variations in knowledge levels.

* Data Collection:

* Questionnaire: The use of a standardized questionnaire to assess knowledge is appropriate. The choice of face-to-face interviews over self-administered questionnaires should be justified, considering potential biases and the need for clarification.

* Language: If the questionnaire was translated into Tigrigna, the rationale for this decision should be explained, particularly given that healthcare professionals in Ethiopia are typically trained in English.

* Questionnaire as Supporting Material: The questionnaire should be included as a supporting document.

* Operational Definitions: Clear operational definitions for "good" and "poor" knowledge should be provided, referencing established standards or guidelines.

Results

The results section should adhere to the journal's specific guidelines for statistical reporting. The underlying dataset should be made publicly available or provided as a supporting document.

Discussion

The discussion section should be more in-depth and focus on the key findings, comparing them to existing literature and providing a critical analysis. The implications of the findings for clinical practice and future research should be discussed.

Limitations and Recommendations

The limitations of the study, including the exclusion of health centers and primary hospitals, should be acknowledged. Recommendations for future research and policy implications should be provided based on the study's findings.

Overall

The manuscript requires significant revision to improve its clarity, coherence, and scientific rigor. Attention to detail, including grammar, punctuation, and language usage, is crucial. A thorough review of the literature and a rigorous analysis of the data are necessary to strengthen the study's contribution to the field.

See the attached file for my point by point comments

Reviewer #2: As i read through the paper it is informative on knowledge of providers related to components of PAC although the criterion used for knowledge was very stringent 100% repose as good versus all less of 100% poor.

The back ground did not mention the Comprehensive Aabortuon Care concept which is another critical milestone on abortion services. The data as well looks a bit older now like 5 years. The discussion section is also abbreviated.

I would say may be is not a kind of strong enough paper to draw attention of PLOS but they still can work on it to address this and other inputs they receive and re-attempt.

6. PLOS authors have the option to publish the peer review history of their article (what does this mean? ). If published, this will include your full peer review and any attached files.

**Do you want your identity to be public for this peer review?** For information about this choice, including consent withdrawal, please see our Privacy Policy .

Reviewer #1: No

Reviewer #2: No

---

## [Editor Report · Decision Letter 1]

1 Apr 2025

Health care providers’ knowledge and associated factors towards the components of post abortion care service in hospitals of Tigray, Northern Ethiopia.

PGPH-D-24-02390R1

Dear Mr Goitom,

We are pleased to inform you that your manuscript 'Health care providers’ knowledge and associated factors towards the components of post abortion care service in hospitals of Tigray, Northern Ethiopia.' has been provisionally accepted for publication in PLOS Global Public Health.

Best regards,

Damen Haile Mariam, MD, MPH, PhD

Academic Editor